# Staffing levels and hospital mortality in England: a national panel study using routinely collected data

Bruna Rubbo ,[1] Christina Saville ,[1,2] Chiara Dall'Ora ,[1,2] Lesley Turner ,[1] Jeremy Jones ,[1] Jane Ball ,[1] David Culliford ,[1,2] Peter Griffiths [1,2]

¹School of Health Sciences, University of Southampton, Southampton, UK
²National Institute for Health Research Applied Research Collaboration (Wessex), University Hospital Southampton, Southampton, UK

**Correspondence to**
Peter Griffiths;
peter.griffiths@soton.ac.uk

## ABSTRACT

**Objectives** Examine the association between multiple clinical staff levels and case-mix adjusted patient mortality in English hospitals. Most studies investigating the association between hospital staffing levels and mortality have focused on single professional groups, in particular nursing. However, single staff group studies might overestimate effects or neglect important contributions to patient safety from other staff groups.

**Design** Retrospective observational study of routinely available data.

**Setting and participants** 138 National Health Service hospital trusts that provided general acute adult services in England between 2015 and 2019.

**Outcome measure** Standardised mortality rates were derived from the Summary Hospital level Mortality Indicator data set, with observed deaths as outcome in our models and expected deaths as offset. Staffing levels were calculated as the ratio of occupied beds per staff group. We developed negative binomial random-effects models with trust as random effects.

**Results** Hospitals with lower levels of medical and allied healthcare professional (AHP) staff (e.g, occupational therapy, physiotherapy, radiography, speech and language therapy) had significantly higher mortality rates (rate ratio: 1.04, 95% CI 1.02 to 1.06, and 1.04, 95% CI 1.02 to 1.06, respectively), while those with lower support staff had lower mortality rates (0.85, 95% CI 0.79 to 0.91 for nurse support, and 1.00, 95% CI 0.99 to 1.00 for AHP support). Estimates of the association between staffing levels and mortality were stronger between-hospitals than within-hospitals, which were not statistically significant in a within–between random effects model.

**Conclusions** In additional to medicine and nursing, AHP staffing levels may influence hospital mortality rates. Considering multiple staff groups simultaneously when examining the association between hospital mortality and clinical staffing levels is crucial.

**Trial registration number** NCT04374812.

## STRENGTHS AND LIMITATIONS OF THIS STUDY

⇒ We collected data from 138 hospitals over 4 years.
⇒ We included numerous clinical staffing groups simultaneously.
⇒ We did not have access to data at the ward level, so our associations were at the hospital level only.
⇒ We used occupied beds-per-full time/contract staff, but this might not be a complete measure of staffing levels, as it does not take into account temporary staff or staff absence.

reducing patients' risk of death and costs.[1–5] Low registered nurse staffing levels are associated with a failure to detect and respond to abnormal physiological vital signs, which is in turn associated with failure to prevent death.[6] This link between registered nurses, failure to detect early deterioration and risk of death has meant the focus of much of the research and associated policy is on nurses. Nonetheless, the healthcare workforce is made up of many different professional groups, who must be mobilised to properly respond to or prevent deterioration and staffing levels of other staff groups are also likely to influence the quality and safety of care.

Studies focussing on the impact of clinical staff other than nurses on hospital mortality rates are relatively scarce. There are several studies on the effect of physician staffing levels, and they all indicate increased risk of patient death with lower medical staffing levels.[7–18] In-hospital patient care is delivered by multidisciplinary staff and no single staff group is solely responsible for patient outcomes. However, only a few studies considered multiple staff groups simultaneously, with some finding partial attenuation or complete absence of the effect observed in models with single staff groups when models were adjusted for other professionally qualified staff.[7 13 14 19] To our knowledge, no recent study has examined the potential effect of allied healthcare professionals (AHP) staffing

## INTRODUCTION

Cost constraints and a quest for safer hospitals put pressure on health systems to find efficient staffing models to meet rising demand for care. Adequate nurse staffing levels in acute hospitals have important implications for positive outcomes for patients, including

levels on clinical outcomes. AHP staff include those in the fields of podiatry and chiropody, dietetics, occupational therapy, orthoptics and optics, physiotherapy, radiography, art, music and drama therapy and speech and language therapy. Only a few studies have focused on hospital pharmacists.[19 20]

In the face of enduring staffing shortages in some professional groups it is important to understand the likely consequences on patient care and to identify priorities. A focus on single professional groups risks unintended consequences through neglect of other important contributions to patient safety and might lead to biased estimates. We therefore aimed to examine the associations between staffing levels of multiple staff groups, and case-mix adjusted patient mortality in hospitals.

## METHODS
### Study design
This was a retrospective observational study using routinely available data on clinical healthcare staffing and hospital mortality.

### Study setting
We included all 138 National Health Service (NHS) hospital trusts providing general acute adult inpatient services in England between April 2015 and March 2019. Hospital trusts are defined as organisational units within the NHS that service a defined geographical area or that provide a specialised function. One trust can therefore encompass several hospitals.

### Data sources and linkage
We linked four data sources that provide trust-level data sets: (1) NHS workforce, which contains detailed information on trust staffing; (2) bed availability and occupancy, which contains data on trust-level available and occupied beds; (3) Estates Returns Information Collection (ERIC), which contains data on trust organisation and structure; and (4) Summary Hospital-level Mortality Indicator (SHMI), which contains data on observed and expected deaths (online supplemental table S1). These data sets are openly available and accessible on the NHS Digital platform, along with the data dictionary.[21 22] We linked the data sets using the unique hospital trust ID (ie, 'Org code').

Standardised mortality rates derived from the SHMI includes all deaths that occur in hospital or within 30-days of discharge among patients admitted to non-specialist acute trusts in the period of a year. SHMI data are derived from the Hospital Episode Statistics (HES) at the level of provider spells (ie, total continuous stay of a patient using a hospital bed at an NHS organisation under the care of one or more consultants, or nursing episode or midwife episode), and the HES-Office of National Statistics linked mortality data (online supplemental table S1).[23] The latter captures deaths that occur outside of hospital.

Data used to calculate the SHMI are submitted by each hospital trust.

SHMI expected deaths are calculated based on individual patient characteristics that can affect the risk of mortality, including the patient's condition for hospitalisation, underlying conditions (Charlson Comorbidity Index),[24] age, gender, method of admission to hospital and year of discharge. Logistic regression models estimate the risk for each provider spell, with binary variable as outcome (ie, died or survived). The SHMI model performs well with early validation accounting for 81% of between-hospital variations.[25] The models are constructed using the preceding 3 years' data, with last year data used to calculate the standardised mortality rates.[26] As SHMI data are published monthly, we used the data set that contains mortality data from April to March the following year to report on annual hospital mortality level for each hospital trust included in this study. Additional information, including data set and reports, can be found at the NHS digital website (https://digital. nhs.uk/data-and-information/publications/ci-hub/ summary-hospital-level-mortality-indicator-shmi).

We obtained hospital staffing data by linking the medical and dental workforce data set (physicians) with the non-medical workforce data set (all other healthcare workers, including nurses, nurse assistants, physiotherapists, pharmacists, among others). Bed occupancy data are published quarterly as averages, with no estimate of monthly variation. For wards open overnight, occupied bed is defined as when it is occupied at midnight on the day in question; for day-only wards, occupied bed is a bed where at least one day case has taken place during the day. Trust teaching status was derived from the ERIC data set, which contains variables on trust profile.

Staffing variables are published monthly. We used full-time equivalent (FTE) values to calculate the annual average available staff at each hospital trust from April to March the following year, to align with the SHMI data. FTE data are standardised measures of the workload that allow for the total workforce workload to be expressed in an equivalent number of full-time staff and are based on the proportion of time each staff are expected to work in a week, which would correspond to an FTE of 1 (eg, 48 hours for doctors, 37.5 hour for nurses and AHP). Overtime and out-of-hour work are not recorded in these data sets.

### Study outcome
The main study outcome was annual standardised all-cause mortality rate derived from observed and expected deaths in the SHMI data set.[27] This is calculated and reported for all trusts providing acute adult services in NHS England.[25] Specialist trusts that do not provide general acute care do not report SHMI and, therefore, were not included in the study.

### Staffing variables
Clinical staff are classified according to occupation codes. We grouped staffing variables by frequency and

occupation as listed by NHS England, merging groups that had mean below 3 with others of a similar occupation in order to create eight groups (ie, medical, surgical, other medical specialties, nurses, support to nurses, AHP, support to AHP and scientific, therapeutic and technical (ST&T) staff). General medicine specialist doctors were grouped with clinical oncology doctors to form the general medicine group, surgeons were grouped with obstetricians and gynaecologists to form the surgical group and all other medical specialties were grouped together to form the other medical group (ie, radiologists, emergency medicine, anaesthetists and pathologists). We excluded psychiatrists and paediatricians. The nurses' group was composed of nurses in adult services only; we did not include nurses from paediatric services. The nurse support group included support to adult and general nurses, and nurses in training. ST&T staff group was composed of multi-therapists, applied psychologist, psychological therapist, pharmacists, dentists, operating theatre staff, social services, and other ST&T staff.

We used the number of general acute occupied beds to calculate the bed-per-staff ratio. Beds assigned to maternity, mental illness and learning disabilities services were excluded; however, general acute beds represented 96.2% of the total beds available across all hospital trusts included in the study. We calculated the average number of occupied beds overnight or day-only per trust per year, and then divided occupied beds per each staff group to obtain the average bed-per-staff levels for each hospital trust per year.

### Hospital-level covariates

Teaching affiliation was coded as yes or no according to the trust type recorded in the ERIC data set. Trust size was calculated based on the number of available general acute beds in each trust.

### Statistical analysis

We initially explored the data using descriptive statistics. Continuous variables were described as mean and SD or median and IQR, depending on the distribution of each variable. Categorical variables were described as frequencies and proportions. Annual mortality rates were calculated by dividing the number of observed deaths by the number of patient spells, across all hospital trusts each year. To explore the relationship between staffing levels and hospital mortality, statistical modelling was conducted at the hospital trust-level using multiple regression models on 4 years of data, with standardised mortality rates regressed onto hospital level staffing levels (expressed as the number of occupied beds-per-staff). We included expected deaths as an offset, as the number of expected deaths is the number of times the event could have happened (ie, the exposure variable).

Multilevel models were adjusted for trust size and affiliation as a teaching hospital, with trust included as a random effect to adjust for clustering. We considered a range of potential models, with model selection based

on theoretical reasoning and model fit (minimising the Akaike information criterion, the Bayesian information criterion and the likelihood ratios (see online supplemental table S2 for alternative frameworks considered)). We report exponentiated coefficients (rate ratios) with 95% CIs for all estimates obtained from the models.

The best performing model (referred to as the main model) was a negative binomial random effects model that included all eight clinical staff groups and hospital characteristics (ie, trust size and teaching status). To explore variations between and within hospital trusts, we also constructed a within–between random effects (WBRE) model, with trusts as random effect, and staffing and hospital characteristics as fixed effects. These models can differentiate between effects arising from staffing differences between hospital trusts and those that are associated with annual changes of staffing levels within each hospital by group-mean centring each trust.[28 29]

We assessed potential collinearity between co-variates using correlation plots and Spearman's correlation coefficient, and the overall multicollinearity for all predictors in the models using generalised variance inflation factors (GVIF). All models had a GVIF <10. As a sensitivity analysis, we re-ran the models after excluding any variables that had a GVIF above 5.[30]

Observations with missing data were removed from the analyses; however, hospital trusts that reported at least 1 year of complete data were included.

We performed data linkage, cleaning, coding and analyses in R statistical software, V.4.0.2 (R Core Team, R Foundation for Statistical Computing, Vienna, Austria), using the lme4 and glmer packages (V.1.1–27.1, Bates *et al*, 2015), and the plm package (.2.4–3, Croissant *et al*, 2021).[31] The full data set is available.[32]

### Patient and public involvement

There was no patient and public involvement in this study.

## RESULTS

### Descriptive statistics

There were 540 observations in the linked data set for the 4-year study period, of which 519 (96%) contained data on all variables and were therefore included in our analyses. There were two hospital trusts that did not report on medical staffing in 2016, with six trusts not reporting these variables from 2017 onwards. For other clinical staff, only one trust did not report staff numbers in 2016 and 2017, and two trusts in 2018, with no missing data in 2019. One trust did not report the number of occupied general acute beds in 2018. The number of hospital trusts that reported on complete data varied, with 137 included in 2016 and 2017, 135 in 2018 and 130 in 2019.

The median number of acute general occupied beds per day was 674.0 (IQR 486.1–911.6). The mean annual observed deaths±SD was 2229±992 (range 525–7468), while the expected deaths were 2228±987.2 (range 665–7591). Annual mortality rates across all hospital trusts

**Table 1** Frequency of occupied bed-per-staff variables across 138 hospital trusts

| Staffing level variables | Median (IQR) | Mean (SD) | SD as % of mean | Mean within-trust SD | Within-trust SD as % of mean |
|---|---|---|---|---|---|
| Bed per medical | 3.66 (3.05–4.30) | 3.70 (0.96) | 25.9 | 0.28 | 7.6 |
| Bed per surgical | 3.79 (3.10–4.28) | 3.73 (0.81) | 21.7 | 0.22 | 5.9 |
| Bed per other medical specialties | 3.96 (3.26–4.69) | 3.94 (1.01) | 25.6 | 0.27 | 6.9 |
| Bed per nurse adult service | 0.61 (0.54–0.68) | 0.61 (0.11) | 18.0 | 0.03 | 4.9 |
| Bed per nurse support | 0.93 (0.80–1.08) | 0.93 (0.21) | 22.6 | 0.07 | 7.5 |
| Bed per AHP | 2.25 (1.88–2.74) | 2.36 (0.89) | 37.7 | 0.18 | 7.6 |
| Bed per AHP support | 10.03 (7.91–13.01) | 11.07 (4.91) | 44.4 | 1.21 | 10.9 |
| Bed per ST&T | 2.24 (1.80–2.82) | 2.31 (0.75) | 32.5 | 0.16 | 6.9 |

AHP, allied healthcare professional; Q1, lower quartile; Q3, upper quartile; ST&T, scientific, therapeutic and technical.

were 3.44% for 2016, 3.46% for 2017, 3.23% for 2018 and 3.31% for 2019. There was a median of 744.2 available general acute beds per trust (IQR 535.7–1009.4, range 143–2704). Hospital trusts in the upper tertile (ie, the upper third of trusts, ranked by size) were classified as large, while those in the lower tertile were deemed small trusts, with the remaining classified as medium-sized hospital trusts. Forty-one (29.7%) hospital trusts were classified as teaching trusts.

Table 1 shows the distribution of staffing variables by their median, IQR, mean and SD across all hospital trusts. There was considerable variation in staffing levels, with nurse staff levels showing least relative variation (SD 18.0% of the mean) and AHP support most (SD 44.4% of the mean). By contrast, variations within trusts as a percentage of the mean were relatively low, with the within trust variation ranging from 4.9% for nurse staffing to 10.9% for AHP support.

All groups of medical staff were highly correlated with each other (range rho=0.68–0.85) (online supplemental figure S1). Nurse staffing levels were strongly correlated with staffing levels in all medical staff groups (rho>0.71). Nurse support staff levels were correlated to registered nurse (RN) (rho=0.30), as were AHP and AHP support (rho=0.47), but neither support staff were correlated to medical staff (0.07 and −0.02, respectively).

### Staffing levels and hospital mortality

Table 2 shows results for the negative binomial random effect models, adjusted for hospital characteristics. In the single staff group models, hospital trusts that had lower staffing levels (ie, more occupied beds per FTE staff) were associated with higher (standardised) mortality rates for all professionally qualified staff groups. The opposite effect was observed for nursing support and AHP support

**Table 2** Exponentiated estimates (rate ratios) for hospital mortality from the multilevel negative binomial random effect models, adjusted for hospital characteristics (n=138)

| Predictors | Estimates for single staff models | 95% CI | Estimates for multiple staff model | 95% CI |
|---|---|---|---|---|
| Bed per medical | 1.05*** | 1.04 to 1.07 | 1.04*** | 1.02 to 1.06 |
| Bed per surgical | 1.04*** | 1.02 to 1.06 | 0.98 | 0.96 to 1.01 |
| Bed per other medical specialties | 1.05*** | 1.03 to 1.06 | 1.03* | 1.00 to 1.06 |
| Bed per nurse adult service | 1.33*** | 1.15 to 1.54 | 1.07 | 0.88 to 1.31 |
| Bed per nurse support | 0.92* | 0.86 to 0.98 | 0.85*** | 0.79 to 0.91 |
| Bed per AHP | 1.02* | 1.00 to 1.04 | 1.04*** | 1.02 to 1.06 |
| Bed per AHP support | 0.99* | 0.99 to 0.99 | 1.00** | 0.99 to 1.00 |
| Bed per ST&T | 1.02** | 1.00 to 1.04 | 0.99 | 0.98 to 1.05 |
| Teaching (reference: not teaching) | 0.96*[#] | 0.92 to 0.99 | 1.01 | 0.98 to 1.05 |
| Medium trust (reference: small trust) | 1.01[#] | 0.97 to 1.05 | 1.01 | 0.98 to 1.04 |
| Large trust (reference: small trust) | 1.00[#] | 0.96 to 1.04 | 1.01 | 0.98 to 1.05 |

[#]Estimates for univariate models, without any staffing variables, * p<0.05, ** p<0.01, *** p<0.001. Estimates are a ratio of observed to expected deaths, all models that included one or multiple staffing levels have been adjusted for hospital characteristics (ie, teaching status and trust size).
AHP, allied healthcare professional; ST&T, scientific, therapeutic and technical.

staff, where hospital trusts with lower staffing levels (ie, more occupied beds per FTE staff) had lower mortality.

In the main model including all staff groups, associations between medical (rate ratio: 1.04, 95% CI 1.02 to 1.06), other medical specialties (1.03, 95% CI 1.00 to 1.06), AHP (1.04, 95% CI 1.02 to 1.06), nurse support (0.85, 95% CI 0.79 to 0.91) and AHP support (1.00, 95% CI 0.99 to 1.00) FTE staffing levels and hospital mortality remained statistically significant, with no change in direction of the effect compared with the single staff models (table 2). Higher mortality rates were observed in hospital trusts with lower levels of medical and AHP staff. In contrast, hospital trusts with lower support staff levels per occupied bed (ie, nursing support and AHP support) had lower mortality rates. The association with RN staffing was attenuated and no longer statistically significant (1.07, 95% CI 0.88 to 1.31), although the observed effect remained relatively large compared with other associations (online supplemental table 3). Similarly, the association between hospital mortality and surgical and ST&T staff, observed in the single staff groups models, were no longer statistically significant in the model adjusted for all staff groups.

Although we noted correlation between staffing variables, our GVIF for all variables was <10. The GVIF for beds per other medical specialties was 6.1 (online supplemental table 3), and therefore over our threshold of 5, so we ran the negative binomial random effects model omitting this variable as a sensitivity analysis and found similar results (not shown; available from authors on request).

### Within-between hospital trust variability

Between-hospital trust effects from the WBRE model were largely the same as those obtained from the main model (ie, between only). For example, the between-rate ratio estimate from WBRE for the FTE medical group was 1.04 (95% CI 1.02 to 1.07), similar to the estimates obtained from the main model (1.04, 95% CI 1.02 to 1.06). However, even for staff groups where between-hospital trusts effects were significant, the within-hospital effects were small and not statistically significant. For example, the within-hospital estimate from the WBRE model for the medical group was 0.99 (95% CI 0.97 to 1.01) (table 3). Despite potentially providing additional information by decomposing the variability into between-estimates and within-estimates, the WBRE model did not improve fit compared with the main model (online supplemental table S2).

### DISCUSSION

After adjustment for hospital characteristics and including multiple clinical staff groups in our models, we found that higher medical and AHP staffing levels were associated with lower mortality rates in acute hospital trusts in England during the period of 2015–2019. In contrast, hospitals with higher support staff per occupied bed had higher mortality rates.

**Table 3** Exponentiated estimates (rate ratios) for hospital mortality from the multilevel within–between negative binomial random effect models, adjusted for hospital characteristics (n=138)

| | Observed deaths | | |
|---|---|---|---|
| **Predictors** | **Incidence rate ratios** | **95% CIs** | **P values** |
| Bed per medical | | | |
| Within-hospital | 0.98 | 0.97 to 1.00 | 0.129 |
| Between-hospital | 1.04 | 1.02 to 1.07 | <0.001 |
| Bed per surgical | | | |
| Within-hospital | 1.03 | 0.99 to 1.06 | 0.106 |
| Between-hospital | 0.98 | 0.95 to 1.01 | 0.222 |
| Bed per other medical specialties | | | |
| Within-hospital | 1.02 | 0.99 to 1.04 | 0.21 |
| Between-hospital | 1.02 | 0.99 to 1.06 | 0.124 |
| Bed per nurse | | | |
| Within-hospital | 0.88 | 0.75 to 1.04 | 0.148 |
| Between-hospital | 1.1 | 0.88 to 1.37 | 0.393 |
| Bed per nurse support | | | |
| Within-hospital | 0.99 | 0.91 to 1.07 | 0.789 |
| Between-hospital | 0.84 | 0.77 to 0.90 | **<0.001** |
| Bed per AHP | | | |
| Within-hospital | 0.99 | 0.96 to 1.01 | 0.296 |
| Between-hospital | 1.04 | 1.02 to 1.07 | **<0.001** |
| Bed per AHP support | | | |
| Within-hospital | 1 | 1.00 to 1.00 | 0.352 |
| Between-hospital | 1 | 0.99 to 1.00 | **0.017** |
| Bed per ST&T | | | |
| Within-hospital | 1.01 | 0.99 to 1.04 | 0.329 |
| Between-hospital | 0.99 | 0.96 to 1.01 | 0.266 |
| Teaching hospital | | | |
| Yes | 1.02 | 0.98 to 1.05 | 0.323 |
| Trust size (reference: small) | | | |
| Medium | 1 | 0.98 to 1.03 | 0.706 |
| Large | 1.01 | 0.98 to 1.04 | 0.381 |

p values in bold are deemed statistically significant (p<0.05)
AHP, allied healthcare professional; ST&T, scientific, therapeutic and technical.

Our study highlights the importance of simultaneously adjusting for multiple clinical staff groups when investigating associations between staffing levels and mortality at the hospital-level. In the single staff group models, significant effects on mortality rates were seen for all staffing groups. However, when adjusted for multiple groups, effects for medical, other medical specialties, AHP, nurse support and AHP support staffing levels remained

statistically significant, but levels of surgical doctors, RN and ST&T staff were no longer significant.

Our findings are in line with a previous study by Jarman *et al*,[7] who found that the association between RN staffing and mortality was no longer significant when adjusting for doctors-per-bed, with higher levels of physician staffing significantly reducing hospital standardised mortality ratios.

Nonetheless, other studies have shown a reduction in hospital mortality when both higher numbers of doctors and nurses per bed were observed.[11 13–15 33–36] The fact that the association between nurse staffing and mortality was not significant in the multivariable models should not be taken to indicate an absence of effect to nursing. Previous studies have reported that much of the variation in nurse staffing occurs between wards within hospitals and within wards over time[1 37] and Keogh noted that hospital nurse staffing levels often bore little relationship to staffing available to be deployed on wards in NHS hospitals,[38] therefore hospital nurse staffing per bed may be a poor indicator of the staffing levels experienced by inpatients. Griffiths *et al*[13] found that when medical staffing levels were accounted for, significant nurse staffing effects were only observed in models using ward-based staffing ratios as opposed to hospital-level staff-per-bed ratios. Longitudinal patient level studies of exposure to variation in nurse staffing confirm that there is an adverse effect of low nurse staffing when measured at this level.[2 39 40]

To our knowledge, this was the first study to include AHP and AHP support staff levels in analysis. Higher levels of AHP staff had a protective effect at the hospital trust level while more AHP support staff were detrimental to patient mortality, a finding that mirrors those obtained for nursing support staff. While causality cannot be assumed for any of these results, there is a potential that previous findings that have emphasised a link between nurse staffing and patient safety could divert attention from the role that other staff groups may play in delivering quality and preventing avoidable deaths.

In our study, high levels of support staff were associated with higher mortality rates. This finding appears to be paradoxical: increasing the number of some staff groups might lead to increased mortality, despite holding all other staffing variables constant. This cannot be fully explored with the given data. One potential reason might be that raising levels of one staff group might affect the distribution of other staff groups to different departments and wards and this could negatively affect mortality. Another possible reason could be that increasing the levels of one staff group might alter the duties of other staff groups— for example, adding more nurse assistants while holding the number of registered nurses constant might increase the workload for the registered nurses as they are now responsible for the supervision of a larger number of nurse assistants, or create an implicit demand for delegation of work that is not suited to the assistants, which could negatively impact mortality.[39] Unfortunately, due to the fact that we do not have granular in-hospital data on deployment of staff and staff responsibility, we are unable to further investigate this, although this mechanism is consistent with evidence on skill mix in the nursing team which shows a tipping point where adding assistants has a positive effect when the number of assistant is low, but a negative one where the number is high.[39]

We found that variations between hospital trusts in the 4-year study period were higher in magnitude than within-hospital trusts variation, and within hospital effects were not statistically significant for any of the clinical staff groups. Structural recruitment and retention difficulties, financial constraints and other non-staffing resources available at each hospital could be main contributors to inter-hospital trust variations. There was relatively little year on year variation in staffing between trusts. Reduced variability of staffing levels within hospitals might be due to the limited number of funded posts in each organisation and could reflect historic staffing levels. Bjerregaard *et al*,[36] using a WBRE model, found staffing levels effects were significant between departments, while within-department variation had no significant effect on hospital mortality. Bjerregaard *et al*[36] proposed that between hospital effects may reflect structural differences in staffing resource between hospitals while the limited within hospital variation we observed could be largely endogenous, reflecting change in staff in response to variation in patient populations and risk that is not reflected in the case-mix adjustment model. This proposition remains untested.

## STRENGTHS AND LIMITATIONS

This was a national study using 4 years of routinely available data. The inclusion of numerous clinical staffing groups simultaneously in the models allowed us to estimate the effect of each staffing level on hospital mortality while considering hospital trust characteristics.

We used a variety of models and selected the one that performed best on our data. Most studies investigating clinical staffing levels and in-hospital mortality used fixed effects models, which only evaluate within-hospital effects. Such models might not appropriately assess the effect of different staffing levels based on variation between hospitals in hospital-level observational studies.

However, our study was limited by the relatively small sample size, as observations were clustered on 138 hospital trusts. We therefore had to group some of the medical specialties and were unable to unpack relatively heterogenous groups such as AHP. Furthermore, we were unable to explore the effects of grade and experience for medical (eg, junior doctors, level of consultants) and RN (eg, senior nurses) staff.

In our study, beds that were occupied by multiple patients in a single day were counted as one single occupied bed. Studies have shown that workload (eg, number of admissions, discharges, additional tasks) in high volume hospitals can lead to higher mortality rates, even when levels of staffing are similar to those observed in low

volume hospitals. Occupied beds-per-staff might not be a good measure of staffing levels, as these do not account for temporary staff or staff absence due to sickness, maternity leave or long-term leave. The extent to which staff are deployed to deliver services such as ambulatory care or other services not delivered to inpatient beds is likely to vary.[41] FTE might not be an accurate representation of actual hours worked by the clinical staff as trusts with similar FTE staff might have a higher or lower rotation of staff responsible for a single bed during a shift, which could affect the quality of care, including mortality rates.[42] Additionally, our bed-to-staff rations might have been overestimated, as we could not include overtime and out-of-hour since these data are not captured in the data sets used for this study.

Future studies should aim to capture different staffing levels responsible for delivering patient care measured at the patient or ward-level, rather than at the hospital-level.

Although SHMI provides good control for patient risk and we controlled for hospital factors and multiple staff groups, the cross-sectional nature of our analysis means that findings need to be interpreted as demonstrating association but not direct evidence of causation.

## CONCLUSION

In conclusion, our study highlights the importance of simultaneously considering multiple staff groups when investigating the effect of clinical staffing levels and mortality at the hospital-level. We showed that the number of AHP and AHP support per occupied beds have a significant impact on patient mortality, yet these groups have largely not been included in previous workforce studies, of which the majority focused exclusively on nursing staff. We also found that hospitals with higher levels of medical staffing had lower mortality rates, while higher levels of support workers were associated with higher hospital mortality.

**Contributors** BR, CS, CD, LT, JJ, JB, DC, and PG made substantial contributions to the conception or design of the work. BR, JJ and PG acquired the data. BR analysed the data. CS and DC independently checked the data analyses. BR, CS, CD, LT, JJ, JB, DC and PG discussed the interpretation of data for the work. BR drafted the work and BR, CS, CD, LT, JJ, JB, DC and PG revised it critically for important intellectual content and gave final approval of the version to be published; PG is guarantor and is accountable for all aspects of the work in ensuring that questions related to the accuracy or integrity of any part of the work are appropriately investigated and resolved.

**Funding** This study/project is funded by the NIHR (Health and Social Care Delivery Research (NIHR128056)). The views expressed are those of the author(s) and not necessarily those of the NIHR or the Department of Health and Social Care.

**Competing interests** All authors have completed the ICMJE uniform disclosure form at http://www.icmje.org/disclosure-of-interest/ and declare: all authors had financial support from the NIHR Health Services and Delivery Research for the submitted work; no financial relationships with any organisations that might have an interest in the submitted work in the previous 3 years; no other relationships or activities that could appear to have influenced the submitted work.

**Patient and public involvement** Patients and/or the public were not involved in the design, or conduct, or reporting, or dissemination plans of this research.

**Patient consent for publication** Not applicable.

**Ethics approval** We did not need ethics approval as this project relied on publicly available routinely collected data.

**Provenance and peer review** Not commissioned; externally peer reviewed.

**Data availability statement** The data set is available from the ePrints repository, DOI: https://doi.org/10.5258/SOTON/D2055.

**ORCID iDs**
Bruna Rubbo http://orcid.org/0000-0002-1629-8601
Christina Saville http://orcid.org/0000-0001-7718-5689
Chiara Dall'Ora http://orcid.org/0000-0002-6858-3535
Lesley Turner http://orcid.org/0000-0003-1489-3471
Jeremy Jones http://orcid.org/0000-0002-2725-0937
Jane Ball http://orcid.org/0000-0002-8655-2994
David Culliford http://orcid.org/0000-0003-1663-0253
Peter Griffiths http://orcid.org/0000-0003-2439-2857

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
