## [Reviewer comments · BMJ Open]

ARTICLE DETAILS

TITLE (PROVISIONAL)	Staffing levels and hospital mortality in England: a national panel study using routinely collected data
AUTHORS	Rubbo, Bruna; Saville, Christina; Dall'Ora, Chiara; Turner, Lesley; Jones, Jeremy; Ball, Jane; Culliford, David; Griffiths, Peter

VERSION 1 – REVIEW

REVIEWER	Vera Winter University of Wuppertal, Schumpeter School of Business and Economics
REVIEW RETURNED	25-Aug-2022

GENERAL COMMENTS	Review on manuscript „ Staffing levels and hospital mortality in England: a national panel study using routinely collected data”, ID bmjopen-2022-066702 The manuscript deals with an interesting, frequently researched topic and uses an interesting data with in my view, two strengths; the differentiation between several HR groups and a longitudinal setting. Yet, there are several aspects which require revision. Introduction To start with, more details on the different staffing groups should be given. More information on the staff, their roles and responsibilities, and their approximate proportion of all staff would be helpful. Then, the presentation of prior research, which currently seems somewhat eclectic, should be aligned and extended. Only one study is mentioned on the effects of physician staffing (in combination with nurse staffing) in the introduction (3, related topics: 8, 9-11, 12). Some more studies then appear in the discussion section (12, 15, 28-33); those should be integrated and summarized in the introduction. Further studies should be searched for to account for their results, e.g., Winter et al. (2021). Studies on skill mix, i.e., assistant nurses, seem to be largely ignored. This should definitively be corrected. The same holds for the other HR groups. Further, I miss theoretical arguments and expectations on how staffing affects mortality. This should account for the fact that mortality is a multi-causal, rather general outcome affected by multiple factors (Kane et al. 2007; Griffiths et al. 2014; Dietermann et al. 2021). In addition, wouldn't one expect more staffing to always be at least as good as less, ceteris paribus? An elaboration on the assumptions and how the staffing groups might relate to each other (e.g., the substitutional nature of RN and assistant nurses and maybe a more complementary nature between physicians and nurses) is essential. Methods I am not familiar with the c-statistic; can you explain its purpose a bit more or omit it completely (“accounting for 81% of between-hospital variations” seems sufficient to me). Why is the SHMI dataset described in study outcome and not data sources?
--

Who calculates SHMI expected deaths? The authors or the persons that provide the SHMI? What models are based on the preceding three years' data?

Again, the study variables subsection starts with description of data set and linkage.

Overtime and out-of-hour-work should be mentioned in limitations.

Data analyses: The choice of regression model and sensitivity analyses seems to be completely data-driven; I would rather like to see a theoretical reasoning (i.e., nature and distribution of dependent variable, nested structure of data, etc.). Moreover, empirical testing should be more than reliance on AIC, BIC, LR test, e.g., a Hausman test or similar for fixed versus random effects model.

Instead of showing the GoF for all kinds of different models, I would prefer to have the regression results of the sensitivity analyses in the appendix.

The authors could try to do separate analyses for within-hospital mortality and 30-days mortality to see whether this makes a difference.

Please add correlations between the independent variables as they provide interesting insights.

Results

Related to my comments in the introduction, I find it difficult to interpret the findings. In my view, the bivariate analyses mainly represent correlations, e.g., one might also argue that hospitals with higher mortality rates employ fewer nurse support staff.

Interpreting the effects of the multivariate analyses is more cumbersome and relates to my question in the introduction: Wouldn't one expect more staffing to always be at least as good as less, *ceteris paribus*? The estimate for one variable represents this variable's effect when keeping all other variables constant; first, the authors should be more explicit and pay more attention to this interpretation. Second, wouldn't we expect the estimates to be all positive? And what do negative estimates then mean / are they really interpretable in the way the authors do?

Maybe, another (theory- and correlation-driven) differentiation would be purposeful; e.g., the effect of (any kind of) nurse staffing as one variable and of nursing skill mix as second variable. This would, in my view, owe to the fact that RN and nurse support staff perform the same tasks to a large extent. At least, using these variables should be presented as sensitivity analyses.

The results of the within-between random effects model are highly interesting, yet also somehow frustrating, as all within-hospital estimates are insignificant. One explanation for this is of course too few within-hospital variation; yet it also calls for acknowledging endogeneity issues and the question on whether the (between-) estimates represent correlation rather than causation. I would recommend the authors to be transparent about these limitations/issues and add a more detailed discussion of this in the discussion section.

I would prefer a table with the full results of the WBRE model instead of a figure.

Discussion

The discussion should be revised based on the comments above.

Studies mentioned should be more aligned with those referred to in the introduction, except they bring up new topics.

I wish the authors good luck with the revision.

References

Dietermann, K., Winter, V., Schneider, U., & Schreyögg, J. (2021). The impact of nurse staffing levels on nursing-sensitive patient outcomes: a multilevel regression approach. *The European Journal of Health Economics*, 22(5), 833-846.

Griffiths, P., Ball, J., Drennan, J., James, L., Jones, J., Recio-

	Saucedo, A., Simon, M.: The association between patient safety outcomes and nurse/healthcare assistant skill mix and staffing levels and factors that may influence staffing requirements (2014) Kane, R.L., Shamliyan, T.A., Mueller, C., Duval, S., Wilt, T.J.: The association of registered nurse staffing levels and patient outcomes: systematic review and meta-analysis. Med. Care 45(12), 1195–1204 (2007). Winter, V., Dietermann, K., Schneider, U., & Schreyögg, J. (2021). Nurse staffing and patient-perceived quality of nursing care: a cross-sectional analysis of survey and administrative data in German hospitals. BMJ open, 11(11), e051133.
--	---

REVIEWER	Hayley B. Gershengorn Montefiore Med Ctr
REVIEW RETURNED	04-Oct-2022

GENERAL COMMENTS	General Comments Rubbo at al. present a study evaluating the association of multidisciplinary staffing levels and mortality for hospitalized patients across 138 hospitals in England in which they find that for most professions, better staffing-to-bed ratios are associated with lower mortality. There is clear value in better understanding the impact of staffing models on patient outcomes; however, I have a number of concerns related to this study as outlined below. Most notably, I worry about the limits inherent to the data itself (major comment #1). At a minimum I think the implications of the data available (and not available) needs to be well addressed in the limitations section; but, honestly, this limitation may be so substantial as to be fatal in my mind. Major Comments  1. Measure of staff levels – Rather than knowing how many staff are onsite (caring for patients) at a given time, what is known is how many staff are contracted to work at the trust. Assuming that all staff work literally their prescribed hours and these hours are the same from trust to trust (within each staff group), the available data may be a reasonable surrogate for the desired data (# of staff onsite to cover the available beds at any given time). This consistency may be true at NHS hospitals, but is definitely not seen in other settings (ie, US hospitals). However, what is also assumed is that each trust deploys the same staff similarly (ie, out of 10 doctors, 5 go to the same two wards in both trusts versus 3 to one and 7 to the other ward in another trust); this, honestly, seems very unlikely. While these limitations are likely un-addressable, it is necessary that the authors speak explicitly to these points (and other limits inherent in their staffing data) more completely in the discussion and how these issues may impact their results and their interpretation. 2. Abstract clarity – The methods/results presented in the Abstract are confusing to me (see individual comments below). As the abstract must stand alone, I think this section needs significant revision. 3. Interpretation of value of comparing multiple staff groups at once – Please see my minor comment 1 under the Discussion section. Based on this study’s results, I do not think the conclusions that “our study highlights the importance of simultaneously considering multiple staff groups” is fully supported. 4. The discussion section is quite long and would benefit from revisions to shorten it. Minor Comments
--

Abstract

- The reporting of “lower mortality rates” with what are presumed to be odds-ratios that are >1 (“1.04, 95% CI 1.02 to 1.06”) is confusing. Please provide clarity of how these models were structured (ie, was the outcome actually survival and not mortality)? Also, please provide an explanation of what these numbers represent – are they odds-ratios or something else?
- Please clarify what are “medical” and what are “allied health” professionals? I would have assumed the former to be physicians and the latter to be people other than physicians or nurses; however, I cannot imagine that nurses were excluded.
- I had assumed that the unit of observation was the hospital (as the outcome is the standardized mortality rate and not the death/survival of an individual); yet, the use of multilevel models and the statement that “Estimates... than within hospitals...” both suggest the models used patients as the unit of observation. Please explicitly clarify this issue.
- I am also having a lot of trouble understanding the sentence (lines 38-40) referenced above, “Estimates...”. I do not understand what is meant by a “within-between random effects model”, nor am I sure that “estimates” can be “higher in magnitude”—do the authors mean the associations found are stronger or something else?
- The last sentence (lines 43-46, “As the main...”) seems out of place to me. While it is plausible that such structural factors such as retention influence this association, it is equally plausible that other things (e.g., other non-staffing resources available at the hospital) drive this finding. I would favor the authors remove this sentence and use the “word room” to better explain their methods.

Article Summary

- Lines 53-54, “We used occupied beds-per-staff”: Based on the following phrase, I assume the authors mean “beds-per-fulltime/contract staff”; is that so? I think this needs to be made more clear to make the latter part of this bullet point make sense.

Introduction

- Line 61: Please define what is meant by “support workforce”.
- Line 64, “skill mix is diluted”: I don’t think one can dilute a “mix”; perhaps this phrase would make more sense as “skill is diluted” or “overall skill is lessened”?
- Lines 68-69: In the ICU setting, there are numerous other studies which more directly look at physician-to-patient (or bed) ratios which are not referenced here: PMID: 16100139; PMID: 25867907; PMID: 28118657; PMID: 34854939.
- Please clarify what is meant by “allied health professionals” at its first mention (lines 71-72).
- Lines 70-73, “To our knowledge, no recent study has examined the potential effect of AHP staffing levels on clinical outcomes...”: I worry about this statement as there are several studies within the ICU context alone evaluating AHP involvement (although not always as the primary exposure) and processes/outcomes of care (e.g., PMID: 28978488; PMID: 25167081; PMID: 34854939).
- Lines 74-80 could probably be condensed to one short sentence at the end of the prior paragraph as the two sentences are long and a bit redundant.

Methods

- Lines 97-98: Given the comment in the Article Summary that the authors could not account for temporary staff or staff absence, it would be help here to explain what staff the data available is for. Some of this information is available in Table S1; yet, as this is the main exposure, it would be helpful to understand what is included and not included better. For example, assuming this includes only contracted staff, is there data on # of staff or on # of fulltime equivalents, is there data on what services they provide (e.g.,

doctors divided into surgeons and non-surgeons), etc.

- Line 106, “The main study outcome was all-cause mortality, with standardised mortality rates...”: It actually seems that the primary outcome was annual standardized all-cause mortality rate – I would simply state this as, as mentioned above in my comments on the Abstract, saying “all-cause mortality” suggests the unit of observation is the individual patient (which it is not).
- Line 132, “for day-only wards”: Why are there day-only wards? Aren’t these all hospitalized patients on hospital wards (which are, presumably, open day and night)?
- Paragraph from lines 135-140: Wouldn’t this make more sense to have after the first sentence of the prior paragraph (lines 129-130) which describes staffing data?
- Lines 137-139: What defines an FTE for AHPs? (definitions are only provided for doctors and nurses)
- I would strongly recommend moving the definitions of staffing groups (lines 143-145) to a separate section entitled “Exposures” to be placed before the section on “Study outcome” as this is the main exposure being investigated and as it is now, it is mentioned almost in passing a good deal into the Methods explanation. Also, please explain what is meant by “scientific, therapeutic, and technical staff”.
- Lines 154-155 seem to be more Results than Methods. Similarly, the comments on how many trusts were included and why some were excluded (lines 159-166) also seems more like Results than Methods.
- Lines 176-178: I am not understanding this... why are expected deaths used as offsets as SMRI already incorporates the expectedness of death? Also, the exposure variable is the staffing variable, no?
- Lines 187-189, “To explore variations between and within hospital trusts, we also constructed a within-between random effects model (WBRE), with trusts as random effect, and staffing and hospital characteristics as fixed effects.”: How does this model differ from the primary models outlined in lines 179-180?
- Please note that ethics approval was not (likely) needed as all data deidentified.

Results

- Lines 248-249: This information about the GVIF<10 is in the Methods so does not, in my mind, need to be repeated here.

Discussion

- Lines 271-277: While I agree that evaluating groups of staff together can be important, I don’t actually think this study demonstrates this as the authors’ propose. The most important staff are those with clinical responsibilities (ie, I’m not sure what S&T are and they surely matter less). So, the only “lost associations” are surgical doctors and RNs. Re: surgical doctors, it is unclear to my why we would think overall mortality would be affected by them (other than through confounding) to begin with; unless the NHS differs in this way, in hospitals I am used to, surgeons rarely perform basic primary-level care to most admitted patients (they serve as consultants). Therefore, only insofar as their staffing levels track with medical doctors’ staffing levels (which they do herein as per Table 1), it is not clear why they would be expected to impact mortality for the population of hospitalized patients at large. The RNs are a different story, but as noted in the Results, the CIs for this estimate are wide and the point estimate continues to suggest more of a potential association of RN staffing levels with mortality than that of other clinicians (but with lost significance due to model power). To me, therefore, the combining of all staffing groups herein is largely moot.
- Lines 301-305: This is a provocative explanation for the finding that higher nurse support staffing levels are associated with higher mortality – namely, that nurses are, in effect, replaced by nursing support staff. Can the authors provide a figure to show the

	correlation noted in line 222 (“Nurse support staff levels were correlated to RN (rho=0.30)”) to highlight this point?  • Lines 306-307: As noted in the Intro, I think other studies do look at these practitioners although maybe not in the manner done here. • Lines 315-317: As mentioned in my comments in the Abstract, I think there are other potential factors to be considered here which could be discussed. Tables/Figures  • Table S1: Please define “spell” (3rd to last row) in a footnote. • Table 1: Suggest condensing columns 2-4 into one column with “median (IQR)” and columns 5 and 6 into one column with “mean (SD)”. • Table 2: Clarify what the “estimates” are... risk ratios? Also, these RRs represent the association of a 1-point change in # of beds-per-staff with a 1-point change in outcome (SHMI)—ie, index from 1 to 2? Please explicitly state this in the tables. • Figure on page 16 of the PDF – Please provide this figure #; and, please re-label the rows with more publication-ready text (ie, “ratio_bed_med_within” as “within hospital association of medical staff bed-to-FTE ratio” or something).
--	---

VERSION 1 – AUTHOR RESPONSE

REVIEWER 1.

Review on manuscript „ Staffing levels and hospital mortality in England: a national panel study using routinely collected data”, ID bmjopen-2022-066702

The manuscript deals with an interesting, frequently researched topic and uses an interesting data with in my view, two strengths; the differentiation between several HR groups and a longitudinal setting. Yet, there are several aspects which require revision.

We are pleased to hear the reviewer found the manuscript interesting and that it added new relevant data to the existing literature.

Introduction

To start with, more details on the different staffing groups should be given. More information on the staff, their roles and responsibilities, and their approximate proportion of all staff would be helpful.

Following the reviewer’s comments, we moved this information from the supplementary material to the main body of the manuscript (lines 150 to 159). We agree that is important for the reader to fully understand the occupational groupings used by the NHS and how we further grouped these to create the 8 staffing groups that we used in our analyses.

Then, the presentation of prior research, which currently seems somewhat eclectic, should be aligned and extended.

Only one study is mentioned on the effects of physician staffing (in combination with nurse staffing) in the introduction (3, related topics: 8, 9-11, 12). Some more studies then appear in the discussion section (12, 15, 28-33); those should be integrated and summarized in the introduction. Further studies should be searched for to account for their results, e.g., Winter et al. (2021).

We have added the studies mentioned in the discussion to the introduction section and expanded on the narrative (lines 71-81).

Studies on skill mix, i.e., assistant nurses, seem to be largely ignored. This should definitively be corrected.

We appreciate the reviewer’s suggestion, and we agree that investigating skill mix is important. The focus of this manuscript was on the combined effect of different staffing groups, so we chose not to highlight studies that looked exclusively at the nursing skill mix or indeed the skill mix within any particular staff group. To an extent, all studies that consider multiple staff groups could be said to be about ‘skill mix’, although the term is most often used when considering issues of labour substitution /

complementarity where the work of different groups are closely coupled (e.g. within the nursing team) we feel it is a potential source of confusion to use it.

The same holds for the other HR groups.

We understand that admin staff are important when it comes to hospital staff configuration, but we decided to limit our study to the staffing groups that have patient-facing clinical duties as the mechanisms share some commonalities (see below).

Further, I miss theoretical arguments and expectations on how staffing affects mortality. This should account for the fact that mortality is a multi-causal, rather general outcome affected by multiple factors (Kane et al. 2007; Griffiths et al. 2014; Dietermann et al. 2021). In addition, wouldn't one expect more staffing to always be at least as good as less, *ceteris paribus*? An elaboration on the assumptions and how the staffing groups might relate to each other (e.g., the substitutional nature of RN and assistant nurses and maybe a more complementary nature between physicians and nurses) is essential.

We have added a theoretical argument to support how staffing influences mortality (lines 60-69). We have elaborated on these issues in the discussion (below) although we do not specifically address complementarity in detail, as this would require modelling interactions between variables which would require complete reanalysis, a larger sample in terms of hospital-trusts, and is unlikely to yield meaningful results when we have no direct information about on team composition and staff deployment. The limited number of cases vs the number of variables forces us to limit the ambition of hypothesis analysis.

Methods

I am not familiar with the c-statistic; can you explain its purpose a bit more or omit it completely ("accounting for 81% of between-hospital variations" seems sufficient to me).

We agree that the addition of c-statistic is not necessary here and have deleted as suggested by the reviewer.

Why is the SHMI dataset described in study outcome and not data sources?

We moved the two paragraphs that detail the SHMI dataset and calculations to the data sources section, as suggested by the reviewer (lines 106-138).

Who calculates SHMI expected deaths? The authors or the persons that provide the SHMI? What models are based on the preceding three years' data?

SHMI data is calculated by NHS Digital. The models are fitted in the SAS software using the LOGISTIC procedure with the model-fitting options RIDGING = ABSOLUTE and NOCHECK. We provided a link to the SHMI NHS digital website, which contains several documents detailing how the SHMI is calculated (lines 123-124).

Again, the study variables subsection starts with description of data set and linkage

We agree with the reviewer's suggestion and therefore moved the paragraphs describing the data linkage and data sources used for staffing variables to the corresponding section, as mentioned above (lines 125-132).

Overtime and out-of-hour-work should be mentioned in limitations.

We have now included this as a limitation in the discussion (lines 368-370).

Data analyses: The choice of regression model and sensitivity analyses seems to be completely data-driven; I would rather like to see a theoretical reasoning (i.e., nature and distribution of dependent variable, nested structure of data, etc.). Moreover, empirical testing should be more than reliance on AIC, BIC, LR test, e.g., a Hausman test or similar for fixed versus random effects model.

Instead of showing the GoF for all kinds of different models, I would prefer to have the regression results of the sensitivity analyses in the appendix.

We agree with the reviewer that the choice of model should be primarily based on theoretical reasoning, and we have now clarified that this was the case for this study (line 184).

The authors could try to do separate analyses for within-hospital mortality and 30-days mortality to see whether this makes a difference.

We are limited to the outcomes provided in the SHMI dataset and while we agree in principle that it would be useful to see both, we are not aware of striking differences in conclusions that arise between related studies (e.g. on nurse staffing) that use different mortality rates (within 30 days of admission vs in hospital only vs within 30 days of discharge vs in hospital only).

Please add correlations between the independent variables as they provide interesting insights.

We thank the reviewer for this suggestion. We have added Figure S1 to the supplementary files, detailing the correlation between all staffing groups included in the models (line 240 and supplementary material).

Results

Related to my comments in the introduction, I find it difficult to interpret the findings. In my view, the bivariate analyses mainly represent correlations, e.g., one might also argue that hospitals with higher mortality rates employ fewer nurse support staff.

We have single staff models, adjusted for hospital characteristics (lines 242-247), and the full models (referred to as main model, lines 252-263), adjusted for all staffing groups detailed in the methods section plus hospital characteristics. We have deleted the word “multivariate” from line 252 as we understand it may cause confusion as all models were multivariate.

Interpreting the effects of the multivariate analyses is more cumbersome and relates to my question in the introduction: Wouldn't one expect more staffing to always be at least as good as less, *ceteris paribus*? The estimate for one variable represents this variable's effect when keeping all other variables constant; first, the authors should be more explicit and pay more attention to this interpretation. Second, wouldn't we expect the estimates to be all positive? And what do negative estimates then mean / are they really interpretable in the way the authors do?

The reviewer raises an important question and considering it does really illustrate why the positive effect of adding more staff cannot simply be taken for granted. In the discussion we have added the following (lines 317-332):

“In our study, higher levels of support staff were associated with higher mortality rates. This finding appears to be paradoxical: increasing the number of some staff groups might lead to increased mortality, when holding all other staffing variables constant. This cannot be fully explored with the given data. One potential reason might be that raising levels of one staff group might affect the distribution of other staff groups to different departments and wards and this could negatively affect mortality. Another possible reason could be that increasing the levels of one staff group might alter the duties of other staff groups – for example, adding more nurse assistants while holding the number of registered nurses constant might increase the workload for the registered nurses as they are now responsible for the supervision of a larger number of nurse assistants, or create an implicit demand for delegation of work that is not suited to the assistants, which could negatively impact mortality.(40) Unfortunately, due to the fact that we do not have granular in-hospital data on deployment of staff and staff responsibility, we are unable to further investigate this, although this mechanism is consistent with evidence on skill mix in the nursing team which shows a tipping point where adding assistants has a positive effect when the number of assistant is low, but a negative one where the number is high (32)”

Maybe, another (theory- and correlation-driven) differentiation would be purposeful; e.g., the effect of (any kind of) nurse staffing as one variable and of nursing skill mix as second variable. This would, in my view, owe to the fact that RN and nurse support staff perform the same tasks to a large extent. At least, using these variables should be presented as sensitivity analyses.

We believe that adding a skill mix variable would not be helpful in this case, as we do not know the distribution of each staff group per ward or department. Staffing group variables are crude measures of staffing levels at each hospital.

The results of the within-between random effects model are highly interesting, yet also somehow frustrating, as all within-hospital estimates are insignificant. One explanation for this is of course too few within-hospital variation; yet it also calls for acknowledging endogeneity issues and the question on whether the (between-) estimates represent correlation rather than causation. I would recommend the authors to be transparent about these limitations/issues and add a more detailed discussion of this in the discussion section.

We have added that the statistically insignificant results from the within-between model could be due to endogeneity issues (lines 342-344), as suggested. We agree with the reviewer that none of our results indicate causation – we have the following statement in our limitations section: “(...) the cross-sectional nature of our analysis means that findings need to be interpreted as demonstrating association but not direct evidence of causation” (lines 372-374).

I would prefer a table with the full results of the WBRE model instead of a figure.

We have amended this accordingly. We have included the following table (Table 3):

Predictors	Incidence Rate Ratios	Observed deaths	
		95% confidence intervals	p-values
Beds per medical			
Within-hospital	0.98	0.97 – 1.00	0.129
Between-hospital	1.04	1.02 – 1.07	<0.001
Beds per surgical			
Within-hospital	1.03	0.99 – 1.06	0.106
Between-hospital	0.98	0.95 – 1.01	0.222
Beds per other medical			
Within-hospital	1.02	0.99 – 1.04	0.21
Between-hospital	1.02	0.99 – 1.06	0.124
Beds per nurse			
Within-hospital	0.88	0.75 – 1.04	0.148
Between-hospital	1.1	0.88 – 1.37	0.393
Beds per nurse support			
Within-hospital	0.99	0.91 – 1.07	0.789
Between-hospital	0.84	0.77 – 0.90	<0.001
Beds per AHP			

Within-hospital	0.99	0.96 – 1.01	0.296
Between-hospital	1.04	1.02 – 1.07	<0.001
Beds per AHP support			
Within-hospital	1	1.00 – 1.00	0.352
Between-hospital	1	0.99 – 1.00	0.017
Beds per ST&T			
Within-hospital	1.01	0.99 – 1.04	0.329
Between-hospital	0.99	0.96 – 1.01	0.266
Teaching hospital			
Yes	1.02	0.98 – 1.05	0.323
Trust size (reference: small)			
Medium	1	0.98 – 1.03	0.706
Large	1.01	0.98 – 1.04	0.381

Discussion

The discussion should be revised based on the comments above. Studies mentioned should be more aligned with those referred to in the introduction, except they bring up new topics.

We have revised our introduction and discussion based on the comments provided by the reviewer – please see highlighted changes in the revised manuscript.

I wish the authors good luck with the revision.

References

Dietermann, K., Winter, V., Schneider, U., & Schreyögg, J. (2021). The impact of nurse staffing levels on nursing-sensitive patient outcomes: a multilevel regression approach. *The European Journal of Health Economics*, 22(5), 833-846.

Griffiths, P., Ball, J., Drennan, J., James, L., Jones, J., Recio-Saucedo, A., Simon, M.: The association between patient safety outcomes and nurse/healthcare assistant skill mix and staffing levels and factors that may influence staffing requirements (2014)

Kane, R.L., Shamliyan, T.A., Mueller, C., Duval, S., Wilt, T.J.: The association of registered nurse staffing levels and patient outcomes: systematic review and meta-analysis. *Med. Care* **45**(12), 1195–1204 (2007).

Winter, V., Dietermann, K., Schneider, U., & Schreyögg, J. (2021). Nurse staffing and patient-perceived quality of nursing care: a cross-sectional analysis of survey and administrative data in German hospitals. *BMJ open*, *11*(11), e051133.

REVIEWER 2.

Comments to the Authors

General Comments

Rubbo et al. present a study evaluating the association of multidisciplinary staffing levels and mortality for hospitalized patients across 138 hospitals in England in which they find that for most professions, better staffing-to-bed ratios are associated with lower mortality. There is clear value in better understanding the impact of staffing models on patient outcomes; however, I have a number of concerns related to this study as outlined below. Most notably, I worry about the limits inherent to the data itself (major comment #1). At a minimum I think the implications of the data available (and not available) needs to be well addressed in the limitations section; but, honestly, this limitation may be so substantial as to be fatal in my mind.

Major Comments

1. Measure of staff levels – Rather than knowing how many staff are onsite (caring for patients) at a given time, what is known is how many staff are contracted to work at the trust. Assuming that all staff work literally their prescribed hours and these hours are the same from trust to trust (within each staff group), the available data may be a reasonable surrogate for the desired data (# of staff onsite to cover the available beds at any given time). This consistency may be true at NHS hospitals, but is definitely not seen in other settings (ie, US hospitals). However, what is also assumed is that each trust deploys the same staff similarly (ie, out of 10 doctors, 5 go to the same two wards in both trusts versus 3 to one and 7 to the other ward in another trust); this, honestly, seems very unlikely. While these limitations are likely un-addressable, it is necessary that the authors speak explicitly to these points (and other limits inherent in their staffing data) more completely in the discussion and how these issues may impact their results and their interpretation.

We agree with the limitations of the data. As the reviewer pointed out, we can only determine the number of staff contracted to work at each hospital trust. However, we believe, despite these limitations, the findings of our study are still pertinent and provide an important insight into the importance of considering all clinical staffing at the hospital-level. We do highlight the limitations of not having access to ward-level data and suggest that these could explain some of our findings, such as the lack of significance for nurse staffing levels on hospital mortality.

We have added the following (lines 298-302): “The fact that the association between nurse staffing and mortality was not associated in the multivariable models should not be taken to indicate an absence of effect to nursing. Previous studies have reported that much of the variation in nurse staffing occurs between wards within hospitals” and other aspects of the discussion (above) now address the limited information about staff deployment

2. Abstract clarity – The methods/results presented in the Abstract are confusing to me (see individual comments below). As the abstract must stand alone, I think this section needs significant revision.

We thank the reviewer for the suggestions, and we have revised the abstract accordingly.

3. Interpretation of value of comparing multiple staff groups at once – Please see my minor comment 1 under the Discussion section. Based on this study’s results, I do not think the conclusions that “our study highlights the importance of simultaneously considering multiple staff groups” is fully supported.

We refer to our reply to comment 1 in the Discussion section.

4. The discussion section is quite long and would benefit from revisions to shorten it.

We have identified redundancies and repetitions in the discussion that we have now removed, cutting more than 200 words from the previous version. However, we have added some new text in response to other comments. We hope the reviewer will agree this is now easier to read.

Minor Comments

Abstract

□ The reporting of “lower mortality rates” with what are presumed to be odds-ratios that are >1 (“1.04, 95% CI 1.02 to 1.06”) is confusing. Please provide clarity of how these models were structured (ie, was the outcome actually survival and not mortality)? Also, please provide an explanation of what these numbers represent – are they odds-ratios or something else?

Our outcome measure is a ratio between the observed deaths and the expected deaths, included in all models as an offset. Therefore, results above 1 indicate a that if you increase the number of occupied beds per one of the staff groups (e.g. medical staff) while holding all other variables included in the model constant, the number of observed deaths will be higher than the number of expected deaths. Results below 1 indicate the opposite.

We thank the reviewer for suggesting that we add more information on how the models were developed. We have amended the abstract accordingly (lines 31-33).

□ Please clarify what are “medical” and what are “allied health” professionals? I would have assumed the former to be physicians and the latter to be people other than physicians or nurses; however, I cannot imagine that nurses were excluded.

The medical staff was composed of physicians specialized in general medicine and clinical oncologists. Allied health professionals include staff in the fields of podiatry and chiropody, dietetics, occupational therapy, orthoptics and optics, physiotherapy, radiography, art, music and drama therapy, and speech and language therapy.

We have moved this information from supplementary materials to the main body of the manuscript (lines 150-159).

□ I had assumed that the unit of observation was the hospital (as the outcome is the standardized mortality rate and not the death/survival of an individual); yet, the use of multilevel models and the statement that “Estimates... than within hospitals...” both suggest the models used patients as the unit of observation. Please explicitly clarify this issue.

The models did use hospital-level data as these were the only data available to us for this study. The within-between model decomposes the total hospital-level variability into the within-hospital component and the between-hospital component (also referred to in the literature as contextual variance, depending on discipline) by group mean centering the staffing variables for each hospital.

□ I am also having a lot of trouble understanding the sentence (lines 38-40) referenced above, “Estimates...”. I do not understand what is meant by a “within-between random effects model”, nor am I sure that “estimates” can be “higher in magnitude”—do the authors mean the associations found are stronger or something else?

The within-between model provides two different estimates:

- a) Within estimates: the de-meaned (i.e. group-mean centered) time-varying variable for each staff group
- b) Between estimates: time-varying variable with the mean of each staff group across all time-points, for each hospital-trust.

The reviewer’s interpretation is correct, and we have revised the mentioned sentence, as suggested: “Estimates of staffing levels on mortality were stronger between- than within-hospitals” (line 40).

□ The last sentence (lines 43-46, “As the main...”) seems out of place to me. While it is plausible that such structural factors such as retention influence this association, it is equally plausible that other things (e.g., other non-staffing resources available at the hospital) drive this finding. I would favor the authors remove this sentence and use the “word room” to better explain their methods.

We have removed this sentence, as suggested by the reviewer.

Article Summary

□ Lines 53-54, “We used occupied beds-per-staff”: Based on the following phrase, I assume the authors mean “beds-per-fulltime/contract staff”; is that so? I think this needs to be made more clear to make the latter part of this bullet point make sense.
The reviewer is correct, and we have amended this sentence accordingly (line 52).

Introduction

□ Line 61: Please define what is meant by “support workforce”.
□ Line 64, “skill mix is diluted”: I don’t think one can dilute a “mix”; perhaps this phrase would make more sense as “skill is diluted” or “overall skill is lessened”?
We agree and have edited this sentence accordingly.

□ Lines 68-69: In the ICU setting, there are numerous other studies which more directly look at physician-to-patient (or bed) ratios which are not referenced here: PMID: 16100139; PMID: 25867907; PMID: 28118657; PMID: 34854939.
We thank the reviewer for suggesting these and we have cited them in the introduction (line 72).

□ Please clarify what is meant by “allied health professionals” at its first mention (lines 71-72). Allied health professionals include staff in the fields of podiatry and chiropody, dietetics, occupational therapy, orthoptics and optics, physiotherapy, radiography, art, music and drama therapy, and speech and language therapy.
We have moved the information detailing the definition of AHP from supplementary to the main body of the manuscript, at first mention as suggested by the reviewer.

□ Lines 70-73, “To our knowledge, no recent study has examined the potential effect of AHP staffing levels on clinical outcomes...”: I worry about this statement as there are several studies within the ICU context alone evaluating AHP involvement (although not always as the primary exposure) and processes/outcomes of care (e.g., PMID: 28978488; PMID: 25167081; PMID: 34854939).
We detailed the definition of AHP in our response above – the studies suggested by the reviewer did not include staffing levels of AHP. Costa et al included respiratory therapists and physical therapists, but not as staffing levels and the outcome they looked at was a process outcome as opposed to a clinical patient outcome.

□ Lines 74-80 could probably be condensed to one short sentence at the end of the prior paragraph as the two sentences are long and a bit redundant.
We have revised this paragraph per the reviewer’s suggestion (lines 70-81).

Methods

□ Lines 97-98: Given the comment in the Article Summary that the authors could not account for temporary staff or staff absence, it would be help here to explain what staff the data available is for. Some of this information is available in Table S1; yet, as this is the main exposure, it would be helpful to understand what is included and not included better. For example, assuming this includes only contracted staff, is there data on # of staff or on # of fulltime equivalents, is there data on what services they provide (e.g., doctors divided into surgeons and non-surgoens), etc. We detailed the healthcare specialties included in each group in supplementary, under the staff variable group section. The medical staff was composed of physicians specialized in general medicine and clinical oncologists. Allied health professionals include staff in the fields of podiatry and chiropody, dietetics, occupational therapy, orthoptics and optics, physiotherapy, radiography, art, music and drama therapy, and speech and language therapy.
We have moved this information to the main body of the manuscript (lines 78-81 and 150-159).

□ Line 106, “The main study outcome was all-cause mortality, with standardised mortality rates...”: It actually seems that the primary outcome was annual standardized all-cause mortality rate – I would simply state this as, as mentioned above in my comments on the Abstract, saying “all-cause mortality” suggests the unit of observation is the individual patient (which it is not).
We have edited this as suggested by the reviewer (line 140).

□ Line 132, “for day-only wards”: Why are there day-only wards? Aren’t these all hospitalized patients on hospital wards (which are, presumably, open day and night)?

Since we don't have ward-level staffing variables, we included both night beds (mainly on wards) and day-only beds as a measure of all beds available at each hospital-trust to calculate the hospital-level bed per staff levels. Our estimates would have been biased if we included hospital-level staffing levels but calculated the ratio of bed per staff by using night ward-beds only.

Paragraph from lines 135-140: Wouldn't this make more sense to have after the first sentence of the prior paragraph (lines 129-130) which describes staffing data?

Lines 137-139: What defines an FTE for AHPs? (definitions are only provided for doctors and nurses)

The definition of FTE for AHP is the same as for nurses – the only group that has a different definition on FTE is the doctors. We have amended this sentence to include AHPs (line 137).

I would strongly recommend moving the definitions of staffing groups (lines 143-145) to a separate section entitled "Exposures" to be placed before the section on "Study outcome" as this is the main exposure being investigated and as it is now, it is mentioned almost in passing a good deal into the Methods explanation. Also, please explain what is meant by "scientific, therapeutic, and technical staff".

This has now been moved from supplementary to the main body of the manuscript, under the Staffing variables subsection in Methods (lines 78-81 and 150-159).

Lines 154-155 seem to be more Results than Methods. Similarly, the comments on how many trusts were included and why some were excluded (lines 159-166) also seems more like Results than Methods.

We have amended this accordingly (lines 212-219, 224-227).

Lines 176-178: I am not understanding this... why are expected deaths used as offsets as SMRI already incorporates the expectedness of death? Also, the exposure variable is the staffing variable, no?

The SHMI incorporates the expectedness of death when calculating the upper and lower band only, for comparison between different trusts. These data were not used in our model, so we had to offset the observed deaths by the expected deaths by producing a ratio of observed by expected deaths per hospital per year.

Lines 187-189, "To explore variations between and within hospital trusts, we also constructed a within-between random effects model (WBRE), with trusts as random effect, and staffing and hospital characteristics as fixed effects.": How does this model differ from the primary models outlined in lines 179-180?

This model uses group mean-centered staffing levels, which allows for the decomposition of the variability into within- and between- hospitals. At level 1, the 'within' effect captures the difference on hospital mortality between units that are higher or lower than average on staffing levels included in the model relative to trusts, whilst at level 2 the 'between' or 'contextual' effect captures the difference between trusts that have higher or lower staffing levels.

Please note that ethics approval was not (likely) needed as all data deidentified.

We have included an ethics statement (lines 403-405).

Result

Lines 248-249: This information about the GVIF<10 is in the Methods so does not, in my mind, need to be repeated here.

We have deleted this accordingly.

Discussion

Lines 271-277: While I agree that evaluating groups of staff together can be important, I don't actually think this study demonstrates this as the authors' propose. The most important staff are those with clinical responsibilities (ie, I'm not sure what S&T are and they surely matter less). So, the only "lost associations" are surgical doctors and RNs. Re: surgical doctors, it is unclear to my why we would think overall mortality would be affected by them (other than through confounding) to begin with; unless the NHS differs in this way, in hospitals I am used to, surgeons rarely perform basic primary-level care to most admitted patients (they serve as consultants). Therefore, only insofar as their staffing levels track with medical doctors' staffing

levels (which they do herein as per Table 1), it is not clear why they would be expected to impact mortality for the population of hospitalized patients at large. The RNs are a different story, but as noted in the Results, the CIs for this estimate are wide and the point estimate continues to suggest more of a potential association of RN staffing levels with mortality than that of other clinicians (but with lost significance due to model power). To me, therefore, the combining of all staffing groups herein is largely moot.

ST&T staff are composed of multi-therapists, applied psychologist, psychological therapist, pharmacists, dentists, operating theatre staff, and social services. While they might have arguably less direct impact on patient mortality, we believe staffing levels of these clinical patient-facing groups could significantly impact patient outcome, and therefore we decided against excluding them in our models.

As for surgeons, our outcome (all-cause standardized mortality) would include patients that were admitted for and had adverse outcome from surgical procedures.

We have moved the definitions of all staff groups included in our study from supplementary to the main text of the manuscript, as we believe this will elucidate the role of staff in each group and therefore highlight their potential impact on hospital mortality (lines 78-81 and 150-159).

Our conclusions that simultaneously adjusting for multiple staff groups are based on the comparison between results from the single staff group models and the multiple staff group model, as explained in by the following: "In the single staff group models, significant effects on mortality rates were seen for all staffing groups. However, when adjusted for multiple groups, effects for medical, other medical specialties, AHP, nurse support, and AHP support staffing levels remained statistically significant, but levels of surgical doctors, RN, and ST&T staff were no longer significant." (lines 288-292).

□ Lines 301-305: This is a provocative explanation for the finding that higher nurse support staffing levels are associated with higher mortality – namely, that nurses are, in effect, replaced by nursing support staff. Can the authors provide a figure to show the correlation noted in line 222 ("Nurse support staff levels were correlated to RN (rho=0.30)") to highlight this point?

We thank the reviewer for this suggestion. We have added Figure S1 to the supplementary files, detailing the correlation between all staffing groups included in the models.

□ Lines 306-307: As noted in the Intro, I think other studies do look at these practitioners although maybe not in the manner done here.
Thank you for bringing these to our attention. As argued above, these studies did not include AHP, as defined by the National Health Service.

□ Lines 315-317: As mentioned in my comments in the Abstract, I think there are other potential factors to be considered here which could be discussed.
We have amended this accordingly.

□
Tables/Figures

□ Table S1: Please define “spell” (3rd to last row) in a footnote.
We have added this to the footnote for Table S1.

□ Table 1: Suggest condensing columns 2-4 into one column with “median (IQR)” and columns 5 and 6 into one column with “mean (SD)”.
We thank the reviewer for the suggested. We have edited the table 1 accordingly (line 235).

□ Table 2: Clarify what the “estimates” are... risk ratios? Also, these RRs represent the association of a 1-point change in # of beds-per-staff with a 1-point change in outcome (SHMI)—ie, index from 1 to 2? Please explicitly state this in the tables.
We have added the following to Table 2’s footnote: “Estimates are a ratio of observed to expected deaths” (line 251).

□ Figure on page 16 of the PDF – Please provide this figure #; and, please re-label the rows with more publication-ready text (ie, “ratio_bed_med_within” as “within hospital association of medical staff bed-to-FTE ratio” or something).
We have changed this into a table per request of the other reviewer. The variables are now clearly labeled and the table is referenced in the text (see Table 3, lines 279-280).

VERSION 2 – REVIEW

REVIEWER	Vera Winter University of Wuppertal, Schumpeter School of Business and Economics
REVIEW RETURNED	01-Dec-2022

GENERAL COMMENTS	Review on revised manuscript „ Staffing levels and hospital mortality in England: a national panel study using routinely collected data”, ID bmjopen-2022-066702.R1 Most of my points have been addressed in the revision. I only have some smaller issues remaining. Data sources The term SHMI is used or both a dataset and a variable/indicator; this is a bit confusing:  • The SHMI includes all deaths that occur in hospital or within 30-days of discharge among 107 patients admitted to non-specialist acute trusts. • The models are 119 based on the preceding three years’ data, with last year data used to calculate the SHMI. The authors should try to be more precise here. The link doesn’t work. Results As the paper’s contribution is to consider multiple staff groups
--

	simultaneously, I think the correlations between the staffing levels deserve some more attention. Right now, the paper only focuses on correlations of nurse staffing, yet the other correlations are also interesting, particularly between a staffing group and an associated support staffing group. Some observations:  - the largest correlations are observed between the different physician staffing groups and between the physician staffing groups with the nurse staffing ($\rho > 0.71$) - the registered nurse levels were moderately correlated with nurse support, AHP, ST&T, and ST&T support staffing levels (ρ between 0.30 and 0.69) - the correlations between a staffing group and its support group (nurses, AHP, ST & T) were 0.30, 0.47, and 0.62 \square thus overall there does not seem to be a substitution effect - nurse support seems unrelated to medical staff and other medical staff, while AHP support was unrelated to medical staff, surgical staff, other medical staff, and nursing staff (ρ between -0.06 and 0.09) In the description of the results of the main model, the estimate for the surgical staff is missing. Discussion I am confused by the sentence: “Between hospital effects may reflect structural differences in staffing resource between hospitals while the limited within hospital variation we observed could be largely endogenous, reflecting variation in patient populations and risk.” Isn’ t the between-variation at risk of endogeneity to a much larger extent than the within-hospital variation?
--	--

REVIEWER	Hayley B. Gershengorn Montefiore Med Ctr
REVIEW RETURNED	11-Nov-2022

GENERAL COMMENTS	General Comments Rubbo et al. have provided thoughtful responses to my previous concerns. I find the Discussion much clearer and crisper and appreciate the added detail for the methods. However, I still have one (previously noted) major concern that I think needs better discussion in the paper. Major Comment I am still concerned that the staffing variables the authors have access to—total number of fulltime staff contracted to work at the trust—greatly limit their ability to address the question they aim to answer. As the authors note in response to this concern, one issue is how the individual trusts staff each ward. The other, more important in my mind, concern, though, is whether an FTE equates to the same number of “days” (or shifts or whatever) covered by the type of clinician. For example, if one trust has 10 fulltime contracted doctors on staff each of whom work 5.2 weeks a year and another has 5 doctors on staff each of whom works 10.4 weeks a year, both trusts will have one doctor present in the trust every day of the year, yet the number of fulltime number of doctors will appear very different (5 vs 10). Perhaps there is a well-adhered to standard at the NHS (which is not present elsewhere) that ensures this variability does not occur. The authors note that “FTE data are based on the proportion of time
--

each staff are expected to work in a week, which would correspond to an FTE of 1 (e.g. 48 137 hours for doctors, 37.5h for nurses and AHP).” However, I know that my FTE states I work 40h/week, yet this bears little resemblance to how many hours I actually work per week (and that’s even before considering that not all of my “FTE” is spent on clinical time). If this is different at the NHS trusts, it needs to be explicitly stated, I think. If, on the other hand, such standardization (not of what is on paper, but what clinicians actually work) is not the case, I am not sure what knowing the total number of contracted FTEs tells us about the patient (or bed)-to-doctor (or other clinician) ratios.

Moreover, the authors note specifically that overtime is not included. However, if a trust is meant to have 5 FTEs of a given clinical type but has only been able to hire 4 (with the rest of the time picked up by overtime), the patients will see the same number of daily clinicians, but the FTE will be reported as 4 instead of a more “true” 5.

This limitation may not be fatal, but I do think a much more nuanced discussion of what impact using this marker of staffing may have on the validity of findings is required. For example, perhaps the finding that having more doctors on staff is associated with lower mortality reflects the fact that while each patient in such a “higher doctor FTE trust” is cared for by a doctor with the same patient load day-after-day, they see a larger number of doctors (because each only works 5.2 instead of 10.4 weeks, from the above example). The authors’ findings would have nothing to do with staffing ratios, per se, therefore; rather, they would speak to issues with care continuity. I think all these types of explanations stemming from the suboptimal staffing data need to be laid out clearly for the reader.

Additional minor comments:

- The authors note in response to one of my prior comments that “Our outcome measure is a ratio between the observed deaths”; however, in the Abstract they say, “with observed deaths as outcome in our models and expected deaths as offset”; and, in the main paper Methods (lines 186-7) they note, “We report exponentiated coefficients (rate ratios) with 95% confidence intervals (CI) for all estimates obtained from the models.”. As such, I remain confused by what the reported values (e.g., in the Abstract: “(1.04, 95%CI 1.02 to 1.06)”) actually denote; usually, when a number is presented for the first time, some explanation of what it is reflects (e.g., “O:E ratio” or “odds-ratio” or “rate-ratio”) is provided. Table 2 suggests these are probably the rate ratios, but it would be helpful to state that clearly both in the Abstract and Results text.
- I appreciate the authors’ explanation of what NHS means by “allied healthcare professional” staff. However, this definition inclusive of people like podiatry/chiropractic, orthoptics and optics, radiography, and art is not the norm; rather this term is used commonly to refer to a specific subgroup of non-doctor, non-nurse clinical staff (e.g., in articles from Australia: PMID: 36346933 & PMID: 36280846; Canada: PMID: 33436453; and the US: PMID: 23102982). As such, if this is the term the authors plan to use, I think, at the least, who it encompasses needs to be clear also in the Abstract as, as it currently reads, this is unintentionally misleading.

VERSION 2 – AUTHOR RESPONSE

Reviewer: 2

Dr. Hayley B. Gershengorn, Montefiore Med Ctr Comments to the Author:
General Comments

Rubbo et al. have provided thoughtful responses to my previous concerns. I find the Discussion much clearer and crisper and appreciate the added detail for the methods. However, I still have one (previously noted) major concern that I think needs better discussion in the paper.

Major Comment

I am still concerned that the staffing variables the authors have access to—total number of fulltime staff contracted to work at the trust—greatly limit their ability to address the question they aim to answer. As the authors note in response to this concern, one issue is how the individual trusts staff each ward. The other, more important in my mind, concern, though, is whether an FTE equates to the same number of “days” (or shifts or whatever) covered by the type of clinician.

For example, if one trust has 10 fulltime contracted doctors on staff each of whom work 5.2 weeks a year and another has 5 doctors on staff each of whom works 10.4 weeks a year, both trusts will have one doctor present in the trust every day of the year, yet the number of fulltime number of doctors will appear very different (5 vs 10). Perhaps there is a well-adhered to standard at the NHS (which is not present elsewhere) that ensures this variability does not occur. The authors note that “FTE data are based on the proportion of time each staff are expected to work in a week, which would correspond to an FTE of 1 (e.g. 48 137 hours for doctors, 37.5h for nurses and AHP).” However, I know that my FTE states I work 40h/week, yet this bears little resemblance to how many hours I actually work per week (and that’s even before considering that not all of my “FTE” is spent on clinical time). If this is different at the NHS trusts, it needs to be explicitly stated, I think. If, on the other hand, such standardization (not of what is on paper, but what clinicians actually work) is not the case, I am not sure what knowing the total number of contracted FTEs tells us about the patient (or bed)-to-doctor (or other clinician) ratios.

We appreciate the reviewer’s comments and concerns over the definition of FTE in the NHS. As defined by NHS digital: “FTE is a standardised measure of the workload of an employed person and allows for the total workforce workload to be expressed in an equivalent number of full-time staff. 1.0 FTE equates to full-time work of 37.5 hours per week, an FTE of 0.5 would equate to 18.75 hours per week.” This definition is available at <https://digital.nhs.uk/data-and-information/publications/statistical/nhs-workforce-statistics/september-2022>, under “key facts”. We have added the following to methods: “FTE data are standardised measures of the workload that allow for the total workforce workload to be expressed in an equivalent number of full-time staff” (page 4, lines 138-140).

We understand the reviewer’s concern over ‘reported vs actually worked’ hours. However, this is an issue for almost all workforce research, as the stated work hours do not necessarily correspond to actual hours worked. We do not believe this should invalidate our results or impede this type of research from being conducted. However, to highlight this limitation, we have added the following to our discussion: “FTE might not be an accurate representation of actual hours worked by the clinical staff as trusts with similar FTE staff might have a higher or lower rotation of staff responsible for a single bed during a shift, which could affect the quality of care, including mortality rates.(42)” (page 11, lines 380-384).

Moreover, the authors note specifically that overtime is not included. However, if a trust is meant to have 5 FTEs of a given clinical type but has only been able to hire 4 (with the rest of the time picked up by overtime), the patients will see the same number of daily clinicians, but the FTE will be reported as 4 instead of a more “true” 5.

We appreciate the reviewer's observation, and we recognise that this is a limitation of using secondary readily available data from NHS digital. We have further emphasised this limitation in our discussion by including the following: "Additionally, our bed-to-staff rations might have been overestimated, as we could not include overtime and out-of-hour since these data are not captured in the datasets used for this study" (page 11, lines 381-382).

This limitation may not be fatal, but I do think a much more nuanced discussion of what impact using this marker of staffing may have on the validity of findings is required. For example, perhaps the finding that having more doctors on staff is associated with lower mortality reflects the fact that while each patient in such a "higher doctor FTE trust" is cared for by a doctor with the same patient load day-after-day, they see a larger number of doctors (because each only works 5.2 instead of 10.4 weeks, from the above example). The authors' findings would have nothing to do with staffing ratios, per se, therefore; rather, they would speak to issues with care continuity. I think all these types of explanations stemming from the suboptimal staffing data need to be laid out clearly for the reader.

We have revised the discussion to include more nuanced analysis of the potential limitations of using FTE and bed-to-staff rations: "trusts with similar FTE staff might have a higher or lower rotation of staff responsible for a single bed during a shift, which could affect the quality of care, including mortality rates.(42)" (please see page 11, lines 378-381 and reply to comments above).

Additional minor comments:

- The authors note in response to one of my prior comments that "Our outcome measure is a ratio between the observed deaths"; however, in the Abstract they say, "with observed deaths as outcome in our models and expected deaths as offset"; and, in the main paper Methods (lines 186-7) they note, "We report exponentiated coefficients (rate ratios) with 95% confidence intervals (CI) for all estimates obtained from the models.". As such, I remain confused by what the reported values (e.g., in the Abstract: "(1.04, 95%CI 1.02 to 1.06)") actually denote; usually, when a number is presented for the first time, some explanation of what it is reflects (e.g., "O:E ratio" or "odds-ratio" or "rate-ratio") is provided. Table 2 suggests these are probably the rate ratios, but it would be helpful to state that clearly both in the Abstract and Results text.

We thank the reviewer for this comment and have amended accordingly by specifying that the estimates are rate ratios (please see page 2 line 38, page 8 lines 256, 260, 280, and page 9 line 289).

- I appreciate the authors' explanation of what NHS means by "allied healthcare professional" staff. However, this definition inclusive of people like podiatry/chiropractic, orthoptics and optics, radiography, and art is not the norm; rather this term is used commonly to refer to a specific subgroup of non-doctor, non-nurse clinical staff (e.g., in articles from Australia: PMID: 36346933 & PMID: 36280846; Canada: PMID: 33436453; and the US: PMID: 23102982). As such, if this is the term the authors plan to use, I think, at the least, who it encompasses needs to be clear also in the Abstract as, as it currently reads, this is unintentionally misleading.

We included the following to the results section of the abstract: "allied healthcare professional (AHP) staff (i.e. podiatry and chiropractic, dietetics, occupational therapy, orthoptics and optics, physiotherapy, radiography, art, music and drama therapy, and speech and language therapy)" (page 2, lines 35-37).

Reviewer: 1

Dr. Vera Winter, University of Wuppertal Comments to the Author:

Review on revised manuscript „ Staffing levels and hospital mortality in England: a national panel study using routinely collected data”, ID bmjopen-2022-066702.R1

Most of my points have been addressed in the revision. I only have some smaller issues remaining.

Data sources

The term SHMI is used for both a dataset and a variable/indicator; this is a bit confusing:

- The SHMI includes all deaths that occur in hospital or within 30-days of discharge among 107 patients admitted to non-specialist acute trusts.
- The models are based on the preceding three years' data, with last year data used to calculate the SHMI.

The authors should try to be more precise here.

We thank the reviewer for drawing our attention to this. We have amended the methods to clarify the difference between the outcome and the dataset by adding the following: “Standardised mortality rates derived from the SHMI includes all deaths that occur in hospital or within 30-days of discharge among patients admitted to non-specialist acute trusts in the period of a year” (page 4, lines 107-109). We have also clarified the following: “The models are constructed using the preceding three years' data, with last year data used to calculate the standardised mortality rates (26)” (page 4, lines 120-122).

The link doesn't work.

We apologise that the link to the data depository does not work. We have been told by the library service of the University of Southampton that the link can only be activated and therefore accessible once the manuscript is published as they cannot assign a DOI to the dataset until the manuscript has been published. However, we have uploaded the full dataset for the reviewer to access prior to the link being activated and available for all to access.

Results

As the paper's contribution is to consider multiple staff groups simultaneously, I think the correlations between the staffing levels deserve some more attention. Right now, the paper only focuses on correlations of nurse staffing, yet the other correlations are also interesting, particularly between a staffing group and an associated support staffing group. Some observations:

- the largest correlations are observed between the different physician staffing groups and between the physician staffing groups with the nurse staffing ($\rho > 0.71$)
- the registered nurse levels were moderately correlated with nurse support, AHP, ST&T, and ST&T support staffing levels (ρ between 0.30 and 0.69)
- the correlations between a staffing group and its support group (nurses, AHP, ST & T) were 0.30, 0.47, and 0.62 → thus overall there does not seem to be a substitution effect
- nurse support seems unrelated to medical staff and other medical staff, while AHP support was unrelated to medical staff, surgical staff, other medical staff, and nursing staff (ρ between -0.06 and 0.09)

We thank the reviewer for their suggestion and have expanded the paragraphs on correlations between staff groups to the following: “All groups of medical staff were highly correlated with each other (range $\rho=0.68$ to 0.85) (Figure S1). Nurse staffing levels were strongly correlated with staffing levels in all medical staff groups ($\rho>0.71$). Nurse support staff levels were correlated to RN ($\rho=0.30$), as were AHP and AHP support ($\rho=0.47$), but neither support staff were correlated to medical staff (0.07 and -0.02, respectively). As the correlations between registered staff and their support were weak to moderate, there does not seem to be a substitution effect between these groups” (page 7, lines 242-248).

In the description of the results of the main model, the estimate for the surgical staff is missing.

We have edited this to include surgical staff: “Similarly, the association between hospital mortality and surgical and ST&T staff, observed in the single staff groups models, were no longer statistically significant in the model adjusted for all staff groups.” (page 8, line 270)

Discussion

I am confused by the sentence: “Between hospital effects may reflect structural differences in staffing resource between hospitals while the limited within hospital variation we observed could be largely endogenous, reflecting variation in patient populations and risk.” Isn’ t the between-variation at risk of endogeneity to a much larger extent than the within-hospital variation?

Thank you for flagging this up. We neglected to properly attribute this proposition and have now amended the sentence to identify the source and the underlying conditions necessary for it to be true. We also identified that it remains a speculation. We have amended as follows (page 11, Lines 353-357):

“Bjerregaard et al.(36) proposed that between hospital effects may reflect structural differences in staffing resource between hospitals while the limited within hospital variation we observed could be largely endogenous, reflecting change in staff in response to variation in patient populations and risk that is not reflected in the case mix adjustment model. This proposition remains untested.”

VERSION 3 – REVIEW

REVIEWER	Vera Winter University of Wuppertal, Schumpeter School of Business and Economics
REVIEW RETURNED	30-Jan-2023
GENERAL COMMENTS	The reviewer completed the checklist but made no further comments.
REVIEWER	Hayley B. Gershengorn Montefiore Med Ctr
REVIEW RETURNED	17-Jan-2023
GENERAL COMMENTS	General Comments -- I appreciate the work Rubbo et al. have done to revise the paper in line with my prior concerns. Major Comments -- none Minor Comment - Table 2: I apologize for not noting this previously, but I’m confused by what the RRs in the first column for teaching status and trust size (medium and large relative to small) are reporting. Presumably each individual staffing model included both of these factors (teaching status and trust size) and, thus, each had its own RR. I would ask the authors to clarify what these single values represent.

VERSION 3 – AUTHOR RESPONSE

Comment from reviewer 2:

Minor Comment

- Table 2: I apologize for not noting this previously, but I'm confused by what the RRs in the first column for teaching status and trust size (medium and large relative to small) are reporting. Presumably each individual staffing model included both of these factors (teaching status and trust size) and, thus, each had its own RR. I would ask the authors to clarify what these single values represent.

The reviewer's interpretation of the estimates for table 2 are correct. We appreciate that the ratios for teaching status and trust size on their own might cause some confusion, therefore we have clarified that all staffing models (single staffing on the left column or multiple staffing on the right column) were adjusted for hospital characteristics; however, the models for the hospital characteristics (i.e. teaching status and trust size) were univariate models, that did not adjust for any staffing variable.

We have edited Table 2 and included the following in the footnote: # estimates for univariate models, without any staffing variables. Estimates are a ratio of observed to expected deaths, all models that included one or multiple staffing levels have been adjusted for hospital characteristics (i.e. teaching status and trust size).